# Vastly extended drug release from poly(pro-17β-estradiol) materials facilitates in vitro neurotrophism and neuroprotection

Anthony R. D'Amato [1], Devan L. Puhl [1,3], Samuel A.T. Ellman[2,3], Bailey Balouch[1], Ryan J. Gilbert [1]* & Edmund F. Palermo [2]*

Central nervous system (CNS) injuries persist for years, and currently there are no therapeutics that can address the complex injury cascade that develops over this time-scale. 17β-estradiol (E2) has broad tropism within the CNS, targeting and inducing beneficial phenotypic changes in myriad cells following injury. To address the unmet need for vastly prolonged E2 release, we report first-generation poly(pro-E2) biomaterial scaffolds that release E2 at nanomolar concentrations over the course of 1–10 years via slow hydrolysis in vitro. As a result of their finely tuned properties, these scaffolds demonstrate the ability to promote and guide neurite extension ex vivo and protect neurons from oxidative stress in vitro. The design and testing of these materials reported herein demonstrate the first step towards next-generation implantable biomaterials with prolonged release and excellent regenerative potential.

[1] Center for Biotechnology and Interdisciplinary Studies, Rensselaer Polytechnic Institute, 110 8th St., Troy, NY 12180, USA. [2] Materials Science and Engineering, Rensselaer Polytechnic Institute, 110 8th St., Troy, NY 12180, USA. [3]These authors contributed equally: Devan L. Puhl, Samuel A.T. Ellman. *email: gilber2@rpi.edu; palere@rpi.edu

njuries to the Central Nervous System (CNS), including Spinal Cord Injury (SCI) and Traumatic Brian Injury (TBI), present some of the most notoriously retractable problems in modern medicine. For example, SCI affects approximately 17,000 patients per year in the USA, with dismally poor clinical outcomes; some extent of tetraplegia or paraplegia occurs in 99.2% of patients[1]. Compelling epidemiological evidence suggests that females have significant SCI recovery advantages, relative to males[2]. Interestingly, the major female sex hormone, 17β-estradiol (E2), exhibits neurotrophic and neuroprotective effects, which might contribute to functional recovery. E2 reduces inflammation[3–5], glial cell reactivity[6], oxidative stress[7–10], and glutamate excitotoxic neuronal death[8,9,11], while also providing neurotrophism in the CNS[12]. Since traumatic CNS injuries persist for years after the initial insult, there is an urgent unmet need for contact guidance scaffolds capable of releasing neuroprotective and regenerative drugs for this long duration[13,14].

Pro-drugs are chemical derivatives of bioactive molecules designed to improve a property of pharmacological importance (e.g. absorption, biodistribution, metabolism and excretion), which are metabolized into the active drug species[15–21]. Indeed, E2 is commonly administered as a pro-drug clinically (e.g. Delestrogen®, Depo®-Estradiol, Estradurin®). Polymerized pro-drugs of the polyester and poly(anhydride ester) type have shown great promise as a means to sustain delivery of NSAIDs, antiseptics, opioids, antioxidants, and antibacterial agents, as pioneered by Uhrich and co-workers[22–28]. Additionally, polyprodrug amphiphiles have been used in advanced gene delivery systems[29,30]. To our knowledge there is only one prior literature example of polyprodrug electrospun fibers[31]. In that study, however, high molecular weight poly(DL-lactide-co-glycolide) was added to the electrospinning solution as a carrier polymer to enable electrospinning of the lower molecular weight polyprodrug. In contrast, the current work presents a poly(proestrogen) material that is suitable for electrospinning without the use of additional carrier polymers.

Electrospun polymer fibers are increasingly utilized for their therapeutic potential in various tissue engineering applications[32]. The morphology of electrospun fibers provides cells with nano-to-micro scale topography necessary to bridge injured tissue, thereby enabling faster regeneration along the orientation of the fibers[33]. Electrospun fibers can also be designed to deliver drugs, typically by non-covalently blending the native drug into the electrospinning solution, which allows for passive release profiles via diffusion. In instances where prolonged drug release is required to target persistent injury or degenerative disease, release kinetics are commonly modified by (1) polymer materials selection from known/available supplies[34], (2) creating dual polymer systems via coaxial electrospinning or incorporating particles into fibers[35–37], and (3) immobilizing drug to the fiber surface[38]. However, the timescale of drug release still remains quite restricted. The longest drug release from electrospun fibers, to date, is ~150 days[39] in vitro, whereas many applications would require local delivery on the timescale of years[13,40].

In this work, we reveal a biomaterial scaffold comprised of electrospun fibers and films composed entirely of poly(pro-E2) that release estrogen locally upon hydrolytic degradation in vitro, over timescales of 1–10 years. We synthesized a polycarbonate that contains two repeating units in the alternating copolymer backbone: the pro-drug of E2 as the payload, and oligo(ethylene glycol) (oEG) modifier units for tuning the hydrophobicity and mechanical properties. Bioactive E2 is released upon hydrolysis of the carbonate bonds in aqueous media. The resulting polyprodrug, hereby referred to as P1 (poly(pro-E2-alt-oEG)), was solution-processed into thin films and electrospun fibers. We quantified the bioactivity of these materials using three different neuron culture assays to reveal the multi-faceted capability of the P1 materials to provide neurotrophism and neuroprotection in vitro, and to direct the extension of neurites in vitro (Fig. 1b). Since the biomaterials release drug over unprecedented timescales that are characteristic of the underlying chemistry, these materials possess unique inherent physical properties that imply great potential as next-generation therapeutic candidates. We employed an innovative and tunable approach to yield fibrous scaffolds composed entirely of a polyprodrug material, the longest-lasting drug release profile ever reported for poly(pro-E2), and a polyprodrug material tailored to promote neuroprotection, neurotrophism, and neuron contact guidance in vitro. Future work will encompass application of these materials in animal models to demonstrate the in vivo degradation rates and

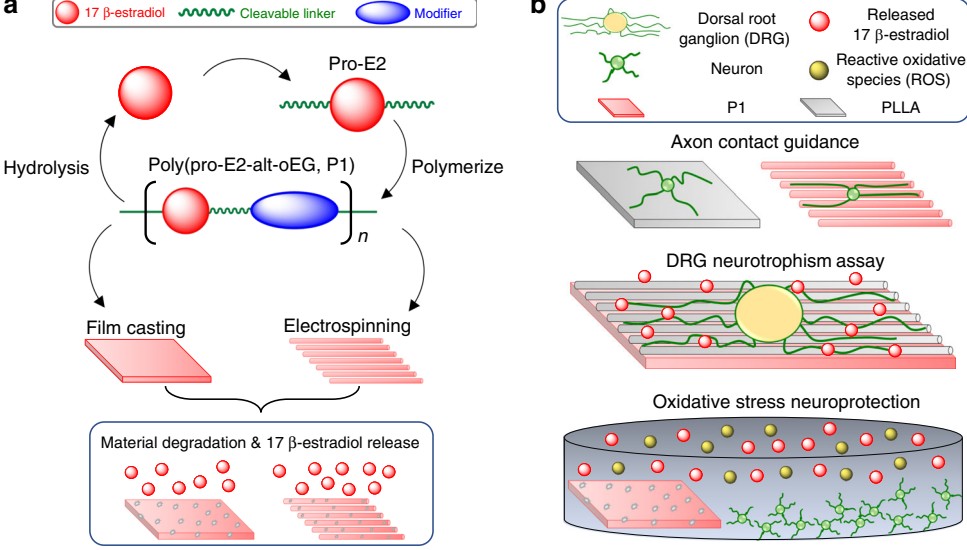

**Fig. 1** P1 material fabrication and testing overview. **a** Step-growth polymerization is used to synthesize P1 from pro-E2 and PEG dithiol monomers to create a copolymer that can be processed into thin films and electrospun fibers that deliver E2 as they degrade via hydrolysis. **b** Three different neuron culture models are used to demonstrate that electrospun P1 fibers provide contact guidance for extending neurites, and P1 films are neurotrophic and neuroprotective against oxidative stress

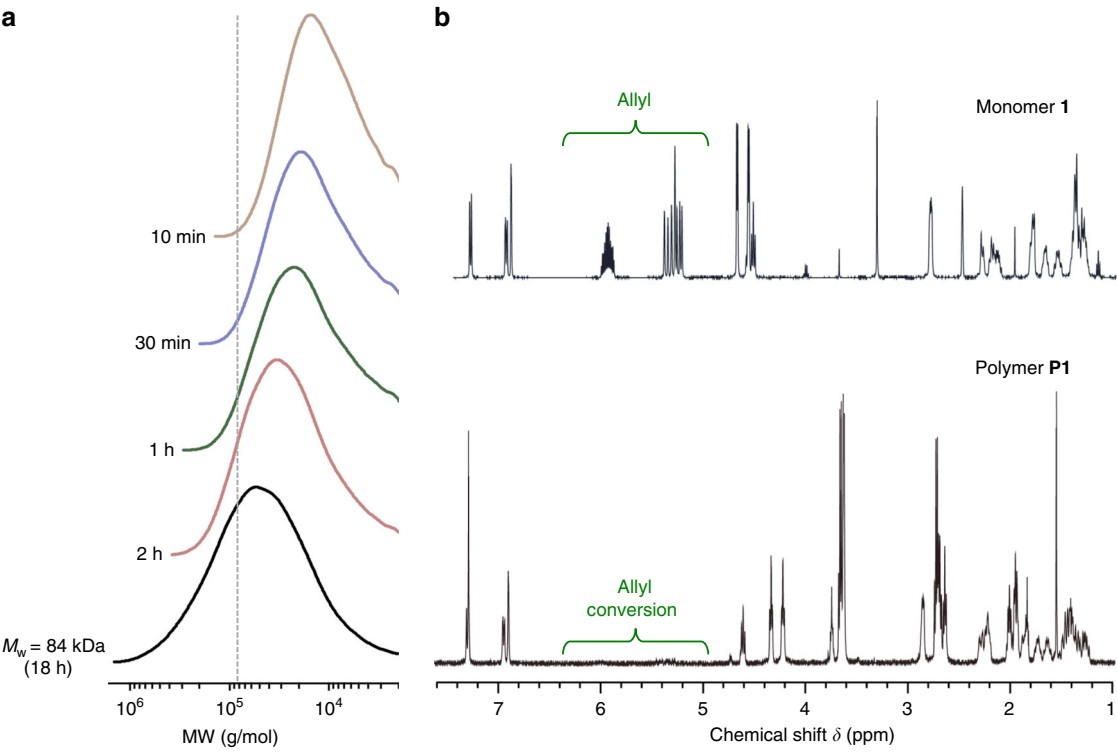

**Fig. 2** Synthesis of P1, a polyprodrug of E2 with hydrolytically cleavable carbonate linkages

**Fig. 3** Validation of successful P1 synthesis. **a** GPC as a function of polymerization time and **b** $^1$H NMR spectra of the monomer 1 and the polymer P1 after 18 h

biocompatibility, as well as SCI recovery models, but these efforts are beyond the scope of our first report on P1 synthesis, characterization, and processing as well as the bioactivity assessment in vitro.

## Results

**Polymer synthesis.** The synthesis and characterization of P1 (Fig. 2) are described in detail in the Supplementary Information (pages S1-7). Briefly, the hydroxyl and phenoxy groups of E2 were functionalized with allyl chloroformate, to yield a small molecule pro-drug of E2 (1) with hydrolytically labile carbonate moieties attached to pendant reactive allyl groups, the latter of which serve as the handle for copolymerization with thiols (Supplementary Fig. 1). Next, step-growth copolymerization with 2,2′-(ethylenedioxy)diethanethiol was performed via photo-initiated thiol-ene radical addition (Supplementary Fig. 2).

Characterization by gel permeation chromatography (GPC, Fig. 3a) confirmed the formation of high MW polymer ($M_w$ = 84 kDa, Đ = 3.73) and the $^1$H and $^{13}$C NMR (Fig. 3b and Supplementary Fig. 3-6) are fully consistent with the proposed chemical structure of the polymer.

**Material properties.** Thermal characterization of P1 by DSC revealed a low glass transition temperature ($T_g$ ~11 °C, Fig. 4a), which is consistent with the soft, rubbery properties of this material at room temperature mainly due to the presence of the flexible oligo(ethylene glycol) linker units in the backbone. This property is atypical of polycarbonates, which are more commonly polymers of stiff/rigid units giving rise to a brittle, glassy solid at room temperature[41]. Thermogravimetric Analysis (TGA) showed decomposition beginning at 270 °C (Fig. 4b), which is typical of a synthetic polycarbonate[41].

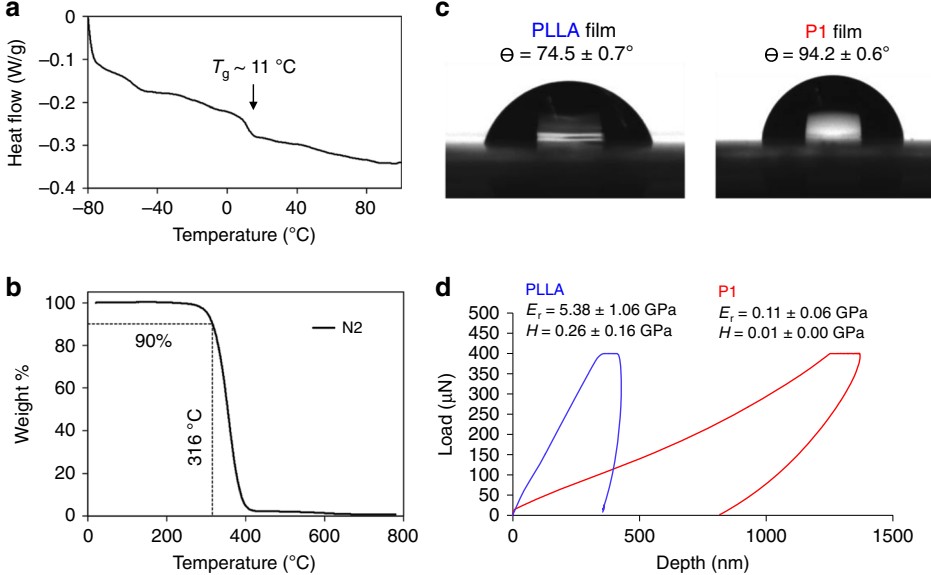

**Fig. 4** Polymer film characterization by **a** DSC, **b** TGA, **c** static water contact angle and **d** nanoindentation

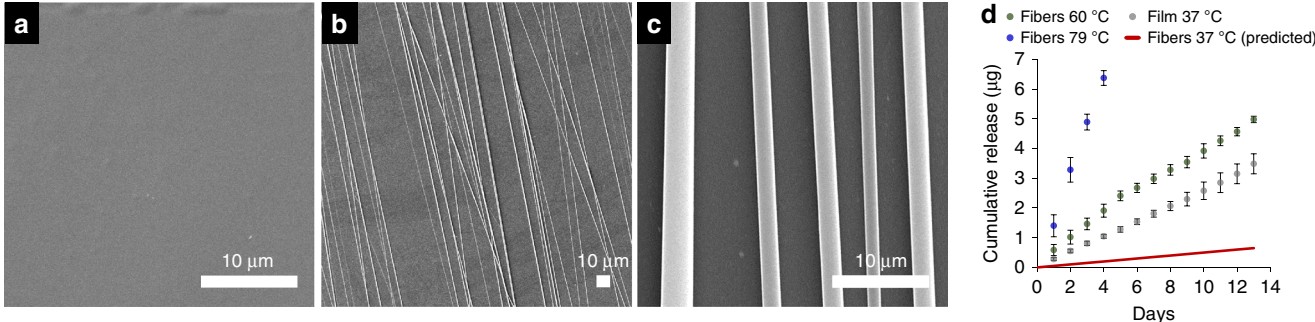

**Fig. 5** P1 material SEM and E2 release kinetics. **a** SEM of P1 films prepared via drop-casting, and **b**, **c** aligned P1 fibers prepared via electrospinning. **d** Cumulative E2 release from P1 fibers and films in PBS ($n \geq 3$, all values are mean ± std. dev., all SEM scale bars are 10 μm)

We also compared the hydrophobicity and mechanical properties of P1 to poly(L-lactic acid) (PLLA), a polyester commonly used in biomaterials[42–44]. P1 films are modestly hydrophobic (94.2 ± 0.6°) compared to PLLA films (74.5 ± 0.7°), according to static water contact angle goniometry images (Fig. 4c). The observed hydrophobicity of P1 films is thus comparable to many common polymeric biomaterials[45]. Nanoindentation (Fig. 4d) demonstrates that the Young's modulus ($E_r$) of P1 films (0.11 ± 0.06 GPa) is approximately 49-fold lower than that of PLLA films (5.38 ± 1.06 GPa), and the hardness (H) of P1 (7 ± 3 MPa) is approximately 37-fold lower than that of PLLA (257 ± 158 MPa).

**Drug release kinetics**. The central goal of this work was to develop biomaterials that consistently deliver physiologically safe and efficacious E2 doses, over very long timescales. The degradation and drug release kinetics for films and microfibers of P1 materials were monitored in vitro by fluorescence spectroscopy (Fig. 5). P1 fiber degradation was studied under accelerated degradation conditions at 60 and 79 °C (Fig. 5d), because the very slow hydrolysis rates at 37 °C rendered the detection of full release impractical on the laboratory timescale. At all temperatures studied, we observed a zero-order release profile (Fig. 5d). By estimating the activation energy from accelerated degradation experiments, we then predicted the release expected at 37 °C from micron-scale electrospun fibers, assuming Arrhenius behavior.

We thus find that P1 electrospun fibers will release 0.21 % of the incorporated E2 in a scaffold per day. This predicted release rate translates to P1 fibers that deliver ~50 ng of E2 daily, corresponding to a ~180 nM concentration in 1 mL of release buffer.

In contrast, thick films of P1 release E2 even more slowly in terms of percentages, at a rate of 0.016% of the incorporated E2 mass in the scaffold per day (equivalent to 266 ± 19 ng/day in terms of mass) at 37 °C. It would thus require as long as 17 years to fully degrade the entire thin film at physiological temperature and pH. Because the thick films contain a much larger absolute quantity of E2, relative to the micron-scale fibers, release of E2 from the films was directly observed and did not require any predictions/assumptions based on accelerated degradation rates.

**Material degradation mechanism**. The difference in drug release rates from fibers and films initially suggested a surface erosion mechanism in vitro (surface/volume dependence). To further interrogate the mechanism of P1 degradation, scanning electron microscopy (SEM) was used to analyze P1 material morphology before and after incubation in aqueous media. After 5 days in neuron culture media at 37 °C, P1 films did not change significantly in morphology (Supplementary Fig. 7), while P1 fiber diameter increased significantly from 3.0 ± 0.1 μm to 5.1 ± 0.4 μm under the same conditions (Supplementary Fig. 8). Both films and fibers were directly observed to swell in water during erosion under accelerated degradation conditions at 80 °C. Under

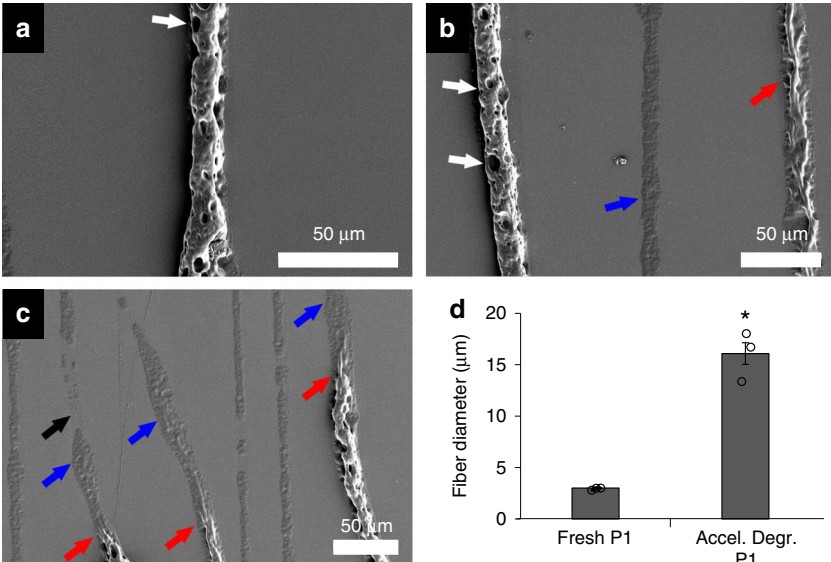

**Fig. 6** Accelerated P1 fiber degradation mechanism in water at 80 °C. **a–c** After 2 days of degradation at 80 °C electrospun P1 fibers exhibited significant morphological changes that suggest bulk erosion. White arrows indicate pitting erosion. Red arrows indicate fibers in late-stage degradation where fibers develop a deflated appearance. Blue arrows indicate fibers near end-stage degradation where a distinct fiber no longer persists, but P1 residue remains. Black arrow indicates an area where a P1 fiber and its remaining residue have completely degraded. **d** A large increase in fiber diameter during the pitting erosion stage further suggests bulk erosion. ($n = 3$, all values are mean ± std. dev., *$p < 0.05$ via Tukey's HSD, all SEM scale bars are 50 μm)

accelerated degradation conditions, P1 films exhibit pronounced pitting and increase significantly in thickness from an initial value of $13.8 \pm 0.5$ μm to a final value of $18.2 \pm 1.7$ μm (Supplementary Fig. 9). Figure 6 shows P1 fibers swelling from an initial average diameter of $3.0 \pm 0.1$ μm to a final average diameter of $16.1 \pm 1.1$ μm after two days of accelerated degradation at 80 °C, which strongly suggests that bulk erosion is at play. Degraded P1 fibers also exhibit clear evidence of spatially heterogenous degradation (e.g. pitting).

**Macrophage-mediated degradation**. We also examined the effects of activated macrophages on P1 film degradation to better mimic the inflammatory environment that exists after CNS injury, as previous studies have shown that these cells can increase the rate of polymer degradation[46]. Accordingly, we cultured activated macrophages directly onto P1 films, and compared these to control P1 films in plain media, and to untreated P1 films (blank as-cast control). Upon 1 week of incubation at 37 °C, there was no significant change in the mass or thickness of the films and SEM images show smooth surfaces without evidence of pitting in all three cases (Supplementary Fig. 10) This is consistent with the proposition that P1 films degrade to release E2 over vastly extended timescales, relative to diffusive release of a small molecule from a polymer matrix. These findings also suggest that activated macrophages do not dramatically accelerate the slow degradation of P1 films, at least not macroscopically over the first 1 week of exposure. Subsequently, we also recovered portions of the P1 films from these experiments and analyzed their molecular weight distributions using GPC, in order to probe degradation at the molecular level (Supplementary Fig. 10f–g). Whereas the untreated and media-treated films showed indistinguishable GPC traces, the sample exposed to activated macrophages developed a slight shoulder peak on the low-MW side of the distribution. This data suggests that chain scission of the polymers in macrophage-treated films did occur to some minor extent, but not substantially enough to manifest as mass loss or thickness reduction. Taken together, these data strongly indicate that the sustained release of E2 from

P1 films is quite likely to persist even in the presence of cells that recapitulate the in vivo environment. We do plan future work to examine the rates of degradation upon subcutaneous implantation in animals, but that subject is beyond the scope of this initial study.

**Fiber contact guidance in vitro**. Electrospun fiber scaffolds are employed in experimental models of neural injury to direct regenerating axons along the orientation of aligned fibers[43,44,47]. To assess the contact guidance potential of this P1 material, we quantified the directed neurite extension in vitro on these fibers. Neurons from dissociated dorsal root ganglion (DRG) were cultured onto P1 fibers electrospun onto glass cover slips. Fluorescence images (Fig. 7) clearly show that neurons which adhere close to P1 fibers exhibit a high propensity to extend their neurites along the fiber direction. In contrast, neurons that adhere to the glass substrate in the absence of fibers extend neurites indiscriminately in all directions. Interestingly, at intermediate adherence distances, neurites extend randomly until they encounter a fiber, at which point they recognize the fiber orientation and begin to extend exclusively along the fiber axis (Supplementary Fig. 11). Although many studies have demonstrated electrospun fiber contact guidance, this is the first to do so with fibers composed entirely of a therapeutic polyprodrug.

**Neurotrophism ex vivo**. To examine the bioactivity of E2 released during P1 degradation and the effects of sustained E2 release on neurons, whole DRG were cultured on P1 films to examine the ability of P1 to promote neurite extension. To analyze P1-derived neurite extension, DRG were cultured onto three types of scaffolds: 1) PLLA fibers electrospun onto a PLLA film (PLLA/PLLA), 2) PLLA fibers electrospun on to a PLLA film with a 100 nM bolus of exogenous E2 at the beginning of culture (PLLA/PLLA + Exo E2), and 3) PLLA fibers electrospun onto a P1 film (PLLA/P1) (Fig. 8). A bolus of E2 was used as a comparison to demonstrate the benefit of sustained E2 release relative to a one-time injection. We used PLLA fibers in this experiment

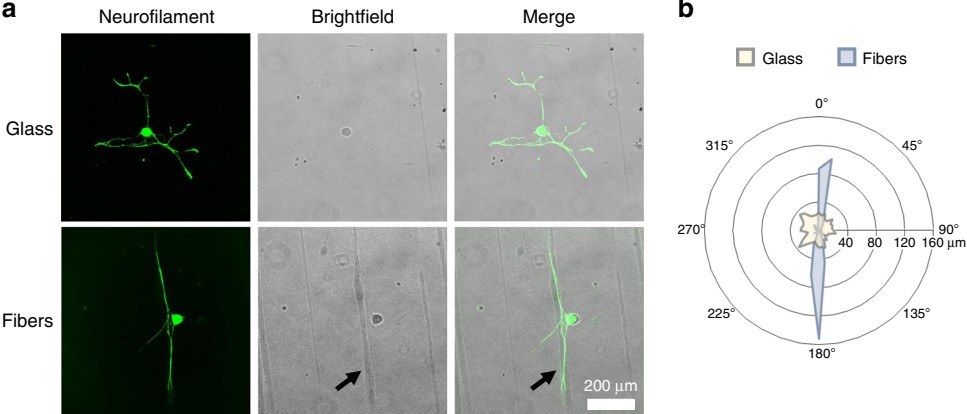

**Fig. 7** Electrospun P1 fiber contact guidance for dissociated cortical neurons. **a** Dissociated cortical neurons cultured onto P1 fibers electrospun onto glass cover slips extended neurites without directional preference when neurons attached to glass surface in the absence of fibers. However, when neurons attached to scaffolds near P1 fibers (as seen in the Brightfield image pane), neurite extension from the neurons followed fiber orientation. **b** Polar histograms representing mean neuron morphology demonstrate the contact guidance cues provided by P1 fibers. ($n = 18$ neurons on glass, and 16 neurons on P1 fibers, scale bar is 200 μm)

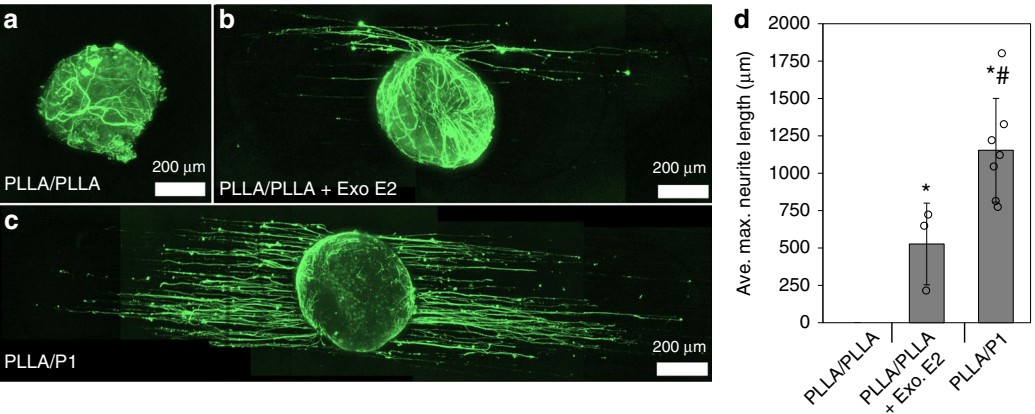

**Fig. 8** P1-derived neurotrophism. **a–c** DRG cultured for four days under each culture condition were fixed and immunostained against neurofilament, **d** and the ten longest neurites from each DRG were measured to yield the value average maximum neurite length, which is the average length of the 10 longest neurites. ($n \geq 3$ for all DRG conditions, all values are mean ± std. dev., *$p < 0.05$ compared to PLLA/PLLA group, #$p < 0.05$ compared to Exo. E2 group via Tukey's HSD, scale bars are 200 μm)

to isolate the effects of E2 released from P1 as the only factor influencing DRG neurite extension.

The average neurite outgrowth from DRG cultured onto PLLA/P1 scaffolds ($1153 \pm 347.0$ μm) was significantly greater than neurite outgrowth from DRG cultured onto PLLA/PLLA scaffolds ($0 \pm 0$ μm, $p < 0.001$, one-way ANOVA) or DRG cultured onto PLLA/PLLA scaffolds with bolus E2 ($526.7 \pm 273.4$ μm, $p = 0.028$, one-way ANOVA). These results show that P1 induces DRG neurite outgrowth that was significantly greater than the neurite outgrowth observed when a one-time bolus of E2 was administered (Fig. 8d).

**Neuroprotection in vitro**. The ability of P1 to protect dissociated cortical neurons from oxidative stress was examined in vitro under hydrogen peroxide ($H_2O_2$) insult. In the absence of peroxide, neuron viability was largely maintained ($91.1 \pm 1.5\%$, Fig. 9). Exposure to $H_2O_2$ (50 μM, 20 min) significantly decreased neuron viability by about half ($46.2 \pm 2.2\%$, $p < 0.0001$, one-way ANOVA) in the absence of P1. Culturing neurons in the presence of a P1 film significantly attenuated oxidative stress-induced neuronal death, showing $83.6 \pm 4.6\%$ viability under the same $H_2O_2$ insult. The viability of neurons cultured with a P1 film and

exposed to $H_2O_2$ was not statistically different than the viability of neurons under control conditions with no insult ($p = 0.604$, one-way ANOVA).

**Biocompatibility with astrocytes**. Having demonstrated the in vitro bioactivity of these materials with neurons, we next studied the biocompatibility of P1 films with other cells of the CNS. In particular, we targeted spinal cord astrocytes because they are the most abundant cell type in the spinal cord and help to maintain homeostasis in the CNS. The compounds released from this material upon degradation are generally considered non-toxic and biocompatible at the relevant concentrations: PEG, $CO_2$, and E2. In order to confirm that P1 films are indeed non-toxic, we cultured spinal cord astrocytes alone or in the presence of P1 films for 7 days and performed a lactate dehydrogenase (LDH) activity assay to quantify the cytotoxicity against astrocytes. We found no evidence of significantly adverse effects, as we expected (Fig. 10).

**Discussion**
The most important outcomes of this study are: (1) We demonstrate the first carbonate-linked polyprodrug formulation

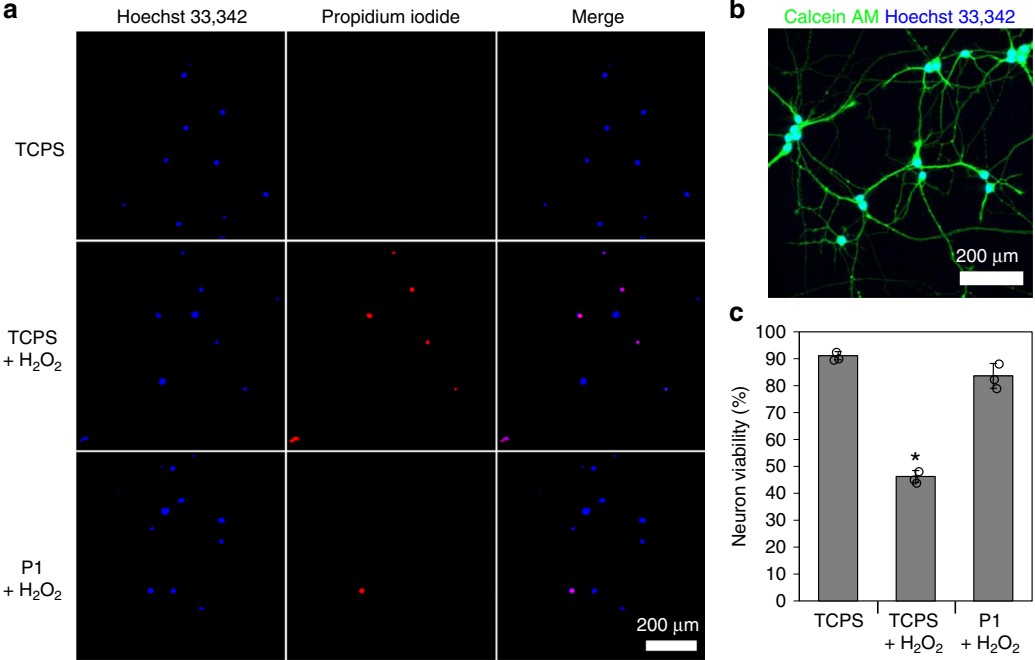

**Fig. 9** P1-derived neuroprotection against $H_2O_2$-induced oxidative stress. **a** Dissociated cortical neurons were cultured onto tissue culture polystyrene (TCPS) with or without a P1 film present in the culture dish. Neurons were insulted with 50 µM $H_2O_2$ and 24 h later neuron viability was assessed via Hoechst 33342 and propidium iodide co-labeling. **b** Calcein labeling was used to demonstrate neuritogenesis for cortical neurons. **c** Neuron viability quantification demonstrates that E2-release from P1 films is neuroprotective against $H_2O_2$-induced oxidative stress. (n = 3 for all culture conditions, neuron viability data are reported as mean ± std. dev., *$p < 0.05$ compared to TCPS control via Tukey's HSD, scale bars 200 µm)

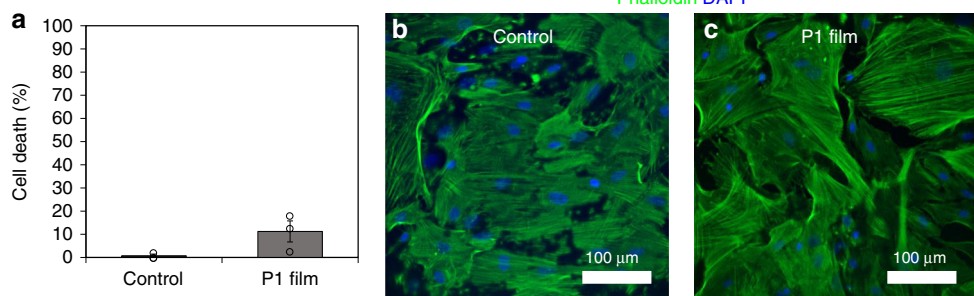

**Fig. 10** P1 film toxicity assessment. **a** Relative levels of LDH were measured to determine percent cell death of spinal cord astrocytes after cultured alone (control) or with P1 films for 7 days. **b**, **c** Phalloidin (green) and DAPI (blue) staining were used to visualize spinal cord astrocytes after being cultured alone or with P1 films for 7 days. (n = 3, all values are mean ± std. dev.)

of E2 and successfully process this polymer into bioactive films and electrospun fibers, (2) these polyprodrug materials display vastly extended E2 release in aqueous buffer and retain their robustness in the presence of activated macrophages, (3) oriented fibrous scaffolds provide contact guidance in vitro, and (4) E2 released during polyprodrug degradation is neurotrophic and neuroprotective against oxidative stress in vitro. Thus, the convergence of polyprodrug click chemistry with the mechanical contact guidance of oriented fiber scaffolds, simultaneously confers directed neurite extension as well as neurotrophic and neuroprotective effects, all within a single, elegant biomaterial. Based on the time scale of E2 release and the results of neuron culture experiments, P1 materials mark an exciting departure from traditional biomaterial drug-delivery approaches that are well established in the literature.

Prior to this work, the longest diffusive E2 release observed from a polymer biomaterial in the literature was achieved from poly(lactic-co-glycolic acid) (PLGA) nanoparticles[48], which released 17.75 µg E2/day over 55 days. Kohane and co-workers

reported polyketal microparticles of pro-E2 that released drug for up to 20 weeks in vitro[49]. According to the E2 release data obtained via P1 hydrolysis in buffer, E2 release from P1 fibers would persist for approximately 1.3 years, while E2 release from thicker P1 films would persist for approximately 17.1 years at physiological temperature and pH. Further, we cultured P1 films with activated macrophages, which are known to increase the rate of polymer degradation. After one week in culture, P1 material degradation was not significantly accelerated by activated macrophages. We thus demonstrated that the slow degradation observed in buffer remains operative in the presence of cells that recapitulate the physiological environment. This appears to be a promising result looking forward to future in vivo applications of P1 materials.

There are many small molecule prodrugs of E2 currently in clinical use for various indications that require prolonged drug release. For example, a single intramuscular injection of estradiol cypionate (Depo®-Estradiol) increased serum E2 levels for 30 days before returning to baseline[50]. Similarly, polyestradiol

phosphate (Estradurin®) is capable of inducing significant increases in E2 serum levels for weeks after administration[51–53]. While we have only studied our P1 material in vitro thus far, it is reasonable to hypothesize that the very high MW of P1, combined with the slow hydrolysis of the polycarbonate structural motif, will likely provide sustained release of E2 in a manner similar to the currently available small molecule pro-drugs, but over much longer timescales.

One of the major breakthroughs herein is that these novel materials as fabricated in this study can deliver from ~50 ng E2/day (fibers) to 260 ng E2/day (films), for orders of magnitude longer timescales than reported previously in the literature. Pharmaceutical applications of E2 in the clinic typically increase serum E2 concentration by 10–150 pg/mL, or 50–750 ng of circulating estrogen in the bloodstream of an average sized adult[50,54]. A fiber mass of approximately 50–750 µg or a film mass of 0.6–9 mg would deliver these amounts of E2 daily and could be sufficient to induce the desired change in E2 serum concentration depending on route of administration and E2 uptake. The key to this extended release is the covalent bonding of the E2 molecule into the backbone of the polycarbonate chains. Carbonates hydrolyze very slowly relative to previously well-studied pro-drug polymer classes[55], such as the poly(ester)s, poly(anhydride)s, poly(ketal)s, and poly(acetal)s[49,56–59], and thus the timescale of release is vastly extended here.

Indeed, the synthesis and chemical composition of P1 are unique in the polymer field: the alternating sequence of rigid and hydrophobic E2 units with flexible and hydrophilic soft oEG units, held together by carbonate linkages, enables a highly tailorable system to precisely control the hydrophobicity of the materials, their mechanical properties, and the kinetics of release. Beyond the scope of this work, we plan to expand upon this canon of polyprodrug-based materials with a systematically tuned library of poly(ester)s, poly(urethane)s, and poly(carbonate)s of pro-drug with varying oEG linker lengths, in order to fully map the vast chemical landscape in this system. We anticipate broad applicability of this class of polyprodrug fibrous scaffolds, well beyond the scope of CNS regeneration explored in this first-generation study.

The large difference in E2 release rates observed between P1 fibers and films suggested that P1 materials may degrade via surface erosion. Findings presented in Fig. 6 and Supplementary Fig. 7–9, however, clearly demonstrate that bulk erosion is the predominant mechanism of P1 material degradation. The slow rate of carbonate hydrolysis relative to fairly modest hydrophobicity is consistent with bulk erosion because the rate of water penetration into the material is expected to well exceed the rate of chemical reaction between water and carbonate bonds. The differences in release rate from P1 fibers versus films are thus likely influenced by either slower diffusion or limited solubility of the hydrolyzed E2 drug (which would also exhibit some extent of surface/volume dependence even in the case of bulk-dominated erosion). The presence of pits in the P1 films and fibers after degradation at 80 °C, is perhaps also related to the SEM sample preparation, during which the water-swollen samples were subjected to high vacuum for sputter coating. Although such pits are typically associated with a surface-limited erosion mechanism, the large degrees of swelling observed here strongly indicate that bulk erosion is operative in this case.

Regarding the P1-derived neurotrophism, it is important to note that DRG culture media in these experiments were not supplemented with exogenous neurotrophic growth factors to stimulate neurite extension. Thus, the observed neurotrophism is ascribed entirely to the release of E2 from the film scaffolds. in vitro, DRG typically require supplementation with neurotrophic growth factors, in order for neurons to survive and extend

neurites. The robust P1-induced neurite extension seen in Fig. 8, despite the absence of any growth factors, demonstrates P1's enormous potential for neural tissue engineering applications. The DRG cultured onto these scaffolds, however, did contain Schwann cells that can produce their own neurotrophic factors in principle (although neurite extension was not observed in the case of the inert control PLLA/PLLA scaffolds). It is not fully understood whether E2 is directly neurotrophic or if E2-derived neurotrophism is mediated by other cells[12]. Multiple studies have observed E2-derived neurotrophism. A review by Spencer et al. suggests that E2 binds to neuronal estrogen receptors and increases expression of the neurotrophin receptors TrkA and TrkB, which bind to the neurotrophins nerve growth factor (NGF) and brain derived neurotrophic factor (BDNF), respectively[60]. A recent study demonstrated that DRG treated with Schwann cell conditioned media (SCM) sprouted more neurites that were significantly longer than DRG neurites grown in un-conditioned media. When SCM-treated DRG were pre-treated with 1 µM E2, the neurotrophic effects of SCM were significantly increased with DRG extending 2.4-fold more neurites with 2.6-fold increased neurite outgrowth distance[61]. DRG that were treated with E2 alone in un-conditioned media also extended neurites that were significantly longer than controls. These findings demonstrated both a direct E2-mediated neurotrophic effect, and indirect E2 neurotrophism mediated by Schwann cells. These prior findings suggest that perhaps the P1-induced neurite growth is the result of both direct and indirect E2-mediated neurotrophism. Importantly, we demonstrated that sustained, long-term release of E2 from polyprodrug scaffolds is much more effective than simply using E2-supplemented media.

Previous studies have established E2-derived neuroprotection against oxidative stress. Some reports suggest that the mechanism of E2 neuroprotection involves modulation of antioxidant enzyme activity, including the expression of superoxide dismutase, catalase, and glutathione peroxidase. By modulating the expression and activity of these enzymes, E2 prevents increased ROS production and subsequent mitochondrial damage[10,62]. E2 also protects against oxidative stress by directly binding to estrogen receptor α (ERα) and increasing the activity of cell survival pathways in neurons. Han et al. previously showed that pre-treatment with E2 activated the MAPK/ERK and PI3K/AKT pathways in neurons, both of which are anti-apoptotic[63]. They further observed significant neuroprotection when pre-treating neurons with E2 (10 nM) prior to $H_2O_2$ insult (250 µM, 30 min), which rescued ~45% of $H_2O_2$-induced neuronal death. The improvement in neuronal viability they observed from bolus E2 administration was similar to the P1-induced improvement in neuronal viability observed here. It is important to note, however, that they pre-treated neurons with E2 15 min before the $H_2O_2$ insult, in contrast to our sustained release approach. This study did not explicitly examine whether the released E2 or the P1 scaffolds themselves are imparting neuroprotection. We find it unlikely that the intact scaffolds are contributing to neuroprotection, however, because the hydroxy and phenoxy groups in E2 are critical to its bioactivity, and these functional groups are masked as carbonates in the polymer backbone. Here, P1-derived neuroprotection combined with contact guidance and neurotrophism demonstrate the tremendous potential of P1 as a promising therapeutic approach to traumatic CNS injury.

In conclusion, this study reveals the first-ever fibrous biomaterial scaffold that is composed entirely of a poly(pro-E2), the first with the ability to give extended drug release on the timescale of multiple years, and the first to direct and promote axonal growth simultaneously via fiber contact guidance, neurotrophism, and neuroprotection. The synthesis and processing of P1 into material scaffolds, such as fibers and films, demonstrate a vertical advance

in neuroprotective and contact guidance scaffolds for neural tissue engineering applications. We show that E2 release from P1 can be sustained, at therapeutically relevant concentrations, for unprecedented timescales on the order of 1–10 years, which is an urgent and previously unmet target in the field of developing biomaterials for drug delivery. Further, we demonstrate that P1 fibers are capable of guiding regenerating cells and tissues, protecting neurons from oxidative stress, and promoting neurite extension. Collectively, the bioactivity and sustained release demonstrated here suggest that these polyprodrug based biomaterials possess an enormous untapped potential to mitigate multiple pathways of CNS injury over timescales that have previously been entirely inaccessible. Whereas this study focused on in vitro drug release and biological activity in terms of in vitro cell response, our future efforts will include the development of animal models to assess the in vivo release of E2 from these scaffolds as well as their efficacy in mitigating the deleterious effects of spinal cord injury.

## Methods

**P1 synthesis**. Detailed synthetic procedures and characterization by $^1H$ and $^{13}C$ NMR, gel permeation chromatography (GPC), and mass spectroscopy, are provided in Supplementary Figs. 1–6.

**Thermal characterization of P1**. We characterized bulk material thermal properties of P1 via differential scanning calorimetry (DSC), and thermogravimetric analysis (TGA). DSC was conducted using a DSC-Q100 (TA Instruments). Raw P1 (5 mg) was hermetically sealed into an aluminum sample pan (TA instruments) and subjected to a 10 °C/min heating/cooling cycle from −80 °C to 200 °C and back to −80 °C, which was repeated twice. The data in Fig. 4 are the final heating scan.

TGA was conducted using a Q50 Thermogravimetric Analyzer (TA Instruments). Raw P1 was placed in an alumina crucible and subjected to a 10 °C/min temperature ramp under $N_2$ up to 800 °C. All DSC and TGA data were analyzed using TA Universal Analysis software.

**Polymer film casting**. PLLA (Cargill Dow LLC) and P1 were dissolved, separately, in chloroform to create separate 5% w/w (w polymer/w solvent) solutions. 50 μL of either solution was then drop-cast onto 15 mm glass cover slips (Knittel Glass) and dried under vacuum overnight. For each polymer, three separate solutions were prepared, and films were drop cast separately from each solution to obtain film materials in technical triplicate (n = 3).

**Contact angle goniometry**. Contact angle goniometry was conducted using a Kruss DSA 100 to characterize P1 and PLLA film hydrophilicity. A 3 μL water drop was placed on either a P1 or PLLA film and the static contact angle was measured using Kruss ADVANCE software.

**Material mechanical characterization**. We characterized P1 film mechanical properties via nanoindentation using a Hysitron TI900 Tribodenter (Bruker). Each polymer film was subjected to 9 indents with a constant load of 400 μN. Indentation depth was recorded with respect to applied load to populate Load-Depth curves and determine material hardness and Young's modulus. Nanomechanical characterization was also conducted on PLLA films to compare mechanical properties between P1 and PLLA films.

**Electrospun fiber fabrication**. P1 fibers were electrospun from a 15% w/w solution of polymer in a 70:30 mixture of dichloromethane and chloroform. Polymer films were affixed to a rotating mandrel (1 cm thick, 22.5 cm in diameter) using a double-sided piece of tape. A syringe filled with polymer solution and affixed to a 22G needle was secured in a variable speed motor syringe pump located directly above the rotating mandrel and the polymer solution was expelled at a steady flow rate of 0.75 mL/hr. The needle was attached to a high voltage power supply using an alligator clamp, and 12.5 kV was applied. Fibers were then electrospun for 15 min over a 5 cm collection distance from the tip of the needle to the rotating mandrel, which rotated at 1200 rpm for the electrospinning duration.

To electrospin PLLA fibers a 12% (w polymer/w solvent) solution was used. This solution was electrospun similarly to the previously described P1 solution, however, the following changes were made to the electrospinning parameters: the PLLA solution was pumped at a steady flow rate of 2 mL/h, 15 kV was applied to the needle, the mandrel was rotated at 1000 rpm, and fibers were collected for 10 min.

Similar to film fabrication, three separate electrospinning solutions were prepared for each polymer (n = 3). Each solution was then electrospun to obtain 10–12 fiber scaffolds from each solution. Each of the scaffolds that were

electrospun from a single solution were considered as belonging to a single technical replicate for statistical analysis purposes.

**Electrospun fiber SEM**. Prior to SEM, fiber scaffolds were mounted on aluminum SEM stubs (Ted Pella) with conductive carbon tape (Ted Pella) and sputter coated with a 0.5 nm Au/Pd coating using a Hummer Technics V sputter coater (Anatech Ltd.). Fiber samples were then imaged using a FEI Versa 3D DualBeam SEM with a low accelerating voltage (2–10 kV) to avoid melting the fibers.

**P1 material degradation kinetics**. To characterize the degradation rate of P1 films, three films were analyzed that were drop casted from three separately prepared P1 solutions (n = 3). P1 films were submerged in 1 mL of PBS in an airtight vial and kept in a cell culture incubator at 37 °C to mimic physiological temperature. Every 24 hours, PBS was collected and stored at −20 °C, then replaced with 1 mL of fresh PBS. A modified version of the P1 film degradation approach was used to characterize P1 fiber degradation since significantly less polymer was present in a P1 fiber scaffold compared to a P1 film. This resulted in lower E2 concentrations that were difficult to detect using standard analytical techniques. Initial attempts to characterize P1 fiber degradation kinetics were performed by submerging scaffolds in PBS at 37 °C. E2 levels in PBS, however, were undetectable when using this approach. Thus, we conducted accelerated degradation studies at two elevated temperatures to compare rate constants to the magnitude of thermal energy using the Arrhenius relation (Eq. 1). We then extracted the activation energy to predict P1 degradation kinetics at 37 °C.

$$\frac{r_1}{r_2} = \frac{e^{\left(\frac{-E}{R*T_1}\right)}}{e^{\left(\frac{-E}{R*T_2}\right)}} \tag{1}$$

To conduct accelerated degradation studies, three fiber scaffolds electrospun from separate P1 solutions (n = 3) were submerged in 1 mL PBS (pH 7.4) in an airtight glass vial and placed in a heating block with aluminum beads held at a steady temperature of either 60 °C or 79 °C. These seemingly arbitrary temperature values were chosen simply because these were the temperatures that the heating block equilibrated to after placing the vials in the aluminum beads. Every 24 h, the vials were removed from the heating block and placed in a 4 °C refrigerator for 15 min to condense any water vapor to ensure that all release buffer was removed at each time point. The PBS was then replaced, and the vials were returned to the heating block. All PBS samples were held at −20 °C until quantifying E2 concentration.

All 1 mL degradation samples were lyophilized, then desalinated by extraction in ethanol. The ethanol was then evaporated off, and the sample raised up in 100 μL of 1-octanol to increase the E2 concentration by a factor of 10 to yield a higher signal during fluorimetric analysis. 1-octanol was chosen as a solvent for this characterization based on a study by Chan et al., which showed that E2 had a high quantum yield in 1-octanol[64]. Prior to fluorimetric analysis, a 2-fold serial dilution of E2 concentrations in 1-octanol ranging from 3 uM to 100 μM was prepared to create a standard curve. A Fluorolog-Tau3 fluorimeter (Horiba Scientific, Edison, NJ) in steady state mode was used to determine the fluorescence of each sample. The value for each sample was then converted to an E2 concentration using the standard curve.

**P1 material degradation mechanism**. The degradation mechanism of P1 fibers and films was studied via SEM. Material scaffolds were submerged in neuron culture media at 37 °C for 5 days to mimic culture conditions, or in water at 80 °C for 2 days to accelerate degradation. Scaffolds were submerged in either condition then imaged via SEM and changes in film thickness or fiber diameter as well as surface pitting were observed as evidence of bulk or surface erosion.

**P1 macrophage-mediated erosion**. Bat et al. showed that in vitro culture with macrophages increased the rate of polymer degradation. Since we showed that our material degrades very slowly in aqueous media, we adapted their protocol to culture our material with macrophages and ensure a more physiologically relevant environment would not negate the prolonged degradation[46].

P1 films were drop casted (n = 3), dried under high vacuum overnight, and the initial mass of all films was measured and recorded. The initial film surface was visualized, and the initial film thickness was determined via SEM.

Murine RAW 264.7 cells were activated in complete DMEM (containing 10% FBS, 1% PenStrep, and 1% GlutaMAX) (Gibco) supplemented with 10 ug/mL LPS (Sigma), and 500,000 cells were seeded onto each P1 film. A separate set of P1 films was submerged in complete DMEM containing 10 ug/mL LPS, but with no cells seeded onto them. All films, with and without cells, were incubated in a 37 °C cell culture incubator. After 24 h, the media was replaced with LPS free complete DMEM and the cells were cultured for a total of 7 days with a media change performed on day 4 of the incubation. Saino et al. showed that 10 ug/mL LPS induces a persistent M1 phenotype in RAW 264.7 cells for at least 7 days[65].

After 7 days, all films were washed with deionized water containing RIPA (Thermo Scientific) to remove salt, protein, and cellular components from the surface of the films. Films were dried under high vacuum overnight, and the final mass of each film was measured and recorded. The final surface was visualized, and the final film thickness was determined via SEM.

**Harvesting DRG**. All animal procedures in this study strictly adhered to the NIH Guidelines for the Care and Use of Laboratory Animals and were approved by the Institutional Animal Care and Use Committee (IACUC) at Rensselaer Polytechnic Institute. For this study, DRG were harvested from neonatal rats (P2). Rats were euthanized via rapid decapitation and spinal columns were isolated and bisected by cutting down the ventral and dorsal aspects using a pair of vannas scissors. DRG were then extracted by clasping the axon bundles projecting on the lateral side of the DRG and carefully lifting the DRG out of the spinal column. Axons were then cut close to the DRG to yield explants that were ready for culture onto material scaffolds. DRG were harvested from multiple animals and cultured onto materials that were fabricated from independently prepared polymer solutions to obtain data in biological and technical triplicate.

**P1-derived neurotrophism**. DRG are typically cultured in cell culture media that contains neurotrophic growth factors to promote neurite outgrowth from the explant. For this set of experiments, however, we did not supplement our culture media with neurotrophic growth factors in order to study the neurotrophic potential of P1. For neurotrophism experiments DRG were cultured in Neurobasal media with 2% B-27 media supplement, 0.5 mM L-Glutamine, and 1% Penicillin/Streptomycin (all media components purchased from Gibco). Immediately after harvesting DRG from neonatal rat spinal columns, DRG were cultured in three different culture conditions. These culture conditions included: (1) DRG cultured onto PLLA fibers electrospun onto PLLA films (PLLA/PLLA), (2) DRG cultured onto PLLA fibers electrospun onto PLLA films with 100 nM exogenous E2 (PLLA/PLLA + Exo E2), and (3) DRG cultured onto PLLA fibers electrospun on to P1 films (PLLA/P1). Prior to DRG culture, both scaffold types were plasma-treated to promote DRG attachment.

After 4 days in culture, DRG were fixed in a 4% paraformaldehyde solution for 15 min. DRG were then washed three times in PBS and incubated overnight in a blocking solution containing 5% bovine serum albumin (BSA, Sigma) and 0.3% Triton-X100 (Sigma) in PBS. After blocking, cells were incubated for 2 h in a solution containing 0.5% BSA, 0.03% Triton-X100, and a 1:250 dilution of the RT97 primary antibody against the neuron-specific intermediate filament protein, neurofilament (DSHB). After primary antibody incubation, DRG were washed in PBS three times then incubated in PBS containing a 1:1000 dilution of AlexaFluor donkey-anti-mouse 488 (Invitrogen) secondary antibody for 2 h. DRG were then washed three final times with PBS and held in fresh PBS at 4 °C until imaging via confocal microscopy.

Immunostained DRG were imaged using an Olympus IX-81 spinning disc confocal microscope. Z-series images of DRG were acquired to account for specimen thickness. Images were then processed into 2D projections using the Z project command on ImageJ.

To compare DRG that were cultured onto different biomaterial scaffolds, we calculated a value termed average neurite length. To calculate average neurite length for each DRG, we used ImageJ software to measure the ten longest neurites in each DRG sample from the base of the DRG body to the tip of each neurite. We then averaged the 10 neurite lengths together to obtain a single average neurite length value for each DRG. This was repeated for each DRG that remained attached to material scaffolds throughout the culture and staining procedures. DRG with longer neurites adhered more strongly to scaffolds, resulting in 3 average neurite length measurements for DRG that were cultured onto PLLA/PLLA scaffolds (with or without exogenous E2), and 7 average neurite length measurements for DRG cultured onto PLLA/P1 scaffolds. Therefore, $n \geq 3$ for P1-derived neurotrophism experiments, with experiments having been performed with at least biological triplicate since DRG were harvested from many different animals to obtain at least 3 attached DRG for each experimental group.

**P1-derived neuroprotection**. Dissociated cortical neuron cultures were purchased from BrainBits LLC and cultured into 12-well tissue culture polystyrene (TCPS) dishes at a seeding density of 2500 cells/cm$^2$ for 5 days, with or without a 15 × 15 mm P1 film present in the well. BrainBits LLC states that dissociated cortical neuron cultures are ~95% pure with an ~5% glial cell contamination. The low neuron seeding density for this experiment was chosen so that individual cells could be imaged via microscopy with a lower likelihood of overlapping cells. Neuroprotection experiments were conducted in technical triplicate by seeding neurons in triplicate wells so that the P1 films present in culture were fabricated using three independently prepared P1 solutions, and the H$_2$O$_2$ insults that neurons were exposed to were prepared independently from a 30% H$_2$O$_2$ (w/w in H$_2$O) stock solution, as purchased from Sigma. The 5-day culture time point was selected based on the BrainBits culture protocol, which states that after 4 days in culture neurons will begin to display axons and dendrites.

On culture day 5, neurons were exposed to a 50 µM H$_2$O$_2$ insult for 20 min. After H$_2$O$_2$ exposure, cells were washed with fresh neurobasal media to ensure complete H$_2$O$_2$ removal, then the media was replaced with fresh neurobasal media. To select the H$_2$O$_2$ insult concentration and duration we first consulted the literature. Studies that use H$_2$O$_2$ to model oxidative stress with neurons in vitro have used a wide range of H$_2$O$_2$ concentrations and treatment durations. For example, Sawada et al. observed approximately 75% of neurons dying after a 10 minute, 30 µM H$_2$O$_2$ insult[9]. In comparison, Gao et al. insulted neurons with 400 µM H$_2$O$_2$ for 2 h and observed 35% death[66]. These inconsistencies in the

literature made it difficult to choose an insult protocol. We experimented with various H$_2$O$_2$ concentrations in the µM range and decided to use the concentration that killed approximately 50% of neurons after 20 min.

After 24 h, neurons were stained with propidium iodide to label dead cells, and Hoechst 33342 to label all nuclei and assess cell viability and determine if P1 scaffolds protected neurons against oxidative stress-induced apoptosis. First, culture media was removed from neurons and cells were rinsed with warm PBS. Neurons were then incubated for 15 min in an imaging solution containing 10 µg/mL Hoechst 33342 (Invitrogen), 2 µg/mL propidium iodide (Invitrogen), 0.025% pluronic F-127 (Sigma), 10 mM HEPES (Sigma), 140 mM NaCl (Sigma), 5 mM KCl (Sigma), 2 mM CaCl$_2$ (Sigma), 2 mM MgCl$_2$ (Sigma), and 5 mM glucose (Sigma) in PBS. After incubation, cells were rinsed once then kept in imaging solution free of Hoechst 33342 and propidium iodide until imaging via confocal microscopy.

Images of dissociated cortical neurons were taken at ×20 magnification and the images corresponding to Hoechst 33342 and propidium iodide fluorescence were overlaid using ImageJ software. Image overlays were analyzed to count total nuclei, as well as nuclei that colocalized with propidium iodide fluorescence, which indicated dead cells. Data were used to determine % viability, a measure of the number of cells that survived after the hydrogen peroxide insult.

**P1 fiber contact guidance**. After validating that P1 films were neurotrophic and neuroprotective in culture, we explored the ability of electrospun P1 fibers to provide contact guidance cues for neurons in culture. Contact guidance is an important aspect of electrospun fiber scaffolds as studies show that this material type can guide regenerating tissue across an injury environment. To analyze P1 fiber-derived contact guidance, we cultured dissociated DRG neurons onto scaffolds with P1 fibers electrospun onto glass. Scaffolds were plasma treated prior to culture to promote cell adhesion. After 12 h in culture, neurons were fixed, immunostained, and imaged following the protocol described previously in the methods section titled P1-derived neurotrophism. Individual neurons were then traced using Neurolucida software (MBF Biosciences) to yield polar histograms that were descriptive of the average length and direction of neurite extension on either fibers or glass.

**Spinal cord isolation**. For this study, spinal cord astrocytes were harvested from neonatal rats (P2). Rats were euthanized and the spinal column was isolated as described previously in the methods section titled Harvesting DRG. The column was cut along the center of the ventral aspect using a pair of vannas scissor. A spinal cord hook was looped under the caudal end of the spinal cord and pulled towards the rostral end to disconnect nerves extending outward from the spinal cord. The spinal cord was isolated from the column and the meninges were removed. Spinal cords were placed in dishes of chilled optiMEM (Gibco) prior to dissociation.

**Spinal cord astrocyte dissociation and culture**. Spinal cords were diced with a scalpel blade (Fine Science Tools) and placed into a 15 mL conical tube (CELL-TREAT). The tube was centrifuged for 5 min at 500 rcf to pellet the tissue. Opti-MEM was removed, and 5 mL of TrypLE Express (Gibco) was added. The spinal cord tissue was resuspended by inverting the tube, and it was placed in a 37 °C cell culture incubator for a total of 10 min. The spinal cord tissue was resuspended every 3-4 min throughout the 10 min incubation. After 10 min, 5 mL of serum containing media was added to the tube and its contents was triturated, using a 5 mL syringe (BD) with a 16-gauge needle (BD), to halt trypsinization. The tissue was then pelleted for 5 min at 500 rcf. The media/TrypLE mixture was removed, and the tissue was resuspended in 10 mL complete DMEM. The cell suspension was added to a T-75 plug seal flask (CELLTREAT) coated for 1 h with 1 µg/mL poly(D-lysine) (Sigma) and cultured in a 37 °C cell culture incubator until the cells were ~80–90% confluent.

Once the cells reached the appropriate confluency, the flask was shaken to remove all non-astrocyte cells. Flasks were first placed in a 37 °C incubated shaker for 1 h and shaken at 200 rpm. Following the 1 h shake, the media was replaced, and the cells were allowed to equilibrate in a cell culture incubator for 3 h. The cells were then placed back in the incubated shaker and shaken at 200 rpm for 18 h[67]. Three separate dissociations were carried out to achieve biological triplicate ($n = 3$).

**P1 film toxicity**. Spinal cord astrocytes were cultured with P1 films to study P1 toxicity in non-neuronal cell populations. P1 toxicity was assessed by comparing the relative levels of LDH in media after spinal cord astrocytes were cultured alone or with P1 films for 7 days. Spinal cord astrocytes were lifted from T-75 flasks using phenol red-free TrypLE Express (Gibco), resuspended in phenol red-free complete DMEM, and seeded into a 24-well culture plate (CELLTREAT) coated, as previously described in the methods section titled Spinal cord astrocyte dissociation and culture, with poly(D-lysine). Cells were seeded at a density of 50,000 cells per well and cultured for 3 days to allow cells to adhere and reach confluency. P1 films, were placed at an angle in each experimental well with the film side facing downward. Media was changed, and all were brought up to a final volume of

2.25 mL to ensure that the film surface was completely submerged. Astrocytes were cultured for 7 days total with no media change.

After 7 days, the media was collected. A LDH activity assay kit (Sigma) and UV/Vis plate reader (Tecan) were used to determine the relative levels of LDH in the each well. All control and experimental groups were normalized to a lysed control well (=100% cell death) and background absorbance from the astrocyte media was subtracted out.

Spinal cord astrocytes were fixed in a 4% paraformaldehyde solution for 10 min. To visualize cell morphology and surface coverage, astrocytes were then incubated in a PBS solution containing 0.1% BSA and a 1:1000 dilution of the filamentous actin stain Phalloidin 488 (Invitrogen) for 2 h. For the last 15 min of the 2 h incubation, a 1:1000 dilution of DAPI (Thermo Scientific) was added to visualize cell nuclei. After the 2 h incubation, cells were washed three times with PBS and stored in fresh PBS at 4 °C until imaging via confocal microscopy. Stained spinal cord astrocytes were imaged and processed as described above (P1-derived neurotrophism).

**Statistical analysis**. All data in this manuscript are presented as mean values ± standard deviation. Statistical comparisons were conducted using JMP software (SAS). Data were compared using a one-way analysis of variance (ANOVA) followed by a post-hoc Tukey's honest significant difference test (HSD) to compare groups directly to each other.

Detailed methods and complete protocols are available in the Supplementary Information.

**Reporting summary**. Further information on research design is available in the Nature Research Reporting Summary linked to this article.

## Data availability

Supporting raw data is available upon request from the corresponding authors. Data in this manuscript are archived in the FigShare repository: https://doi.org/10.6084/m9.figshare.c.4681220.v1

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

## Acknowledgements

We thank Dr. Deniz Rende for assistance with nanoindentation measurements. We acknowledge the following funding sources: NY State Spinal Cord Injury Review Board (NYSCIRB) Predoctoral Fellowship to ARD (C32631GG), NIH R01(NS092754) and NSF CAREER Award (1105125) to R.J.G., and NSF CAREER Award (1653418) and 3 M Non-tenured Faculty Award to E.F.P.

## Author contributions

A.R.D., R.J.G., and E.F.P. conceptualized and designed the study. All experiments were conducted by A.R.D., S.A.T.E., B.B., and D.L.P. A.R.D., R.J.G., D.L.P., and E.F.P. all contributed to the writing and editing of the manuscript.

## Competing interests

A.R.D., S.A.T.E., R.J.G., and E.F.P. are co-inventors on a patent pending (Provisional patent application no. 62746243) related to the polyprodrug synthesis and processing into biomaterials described in this manuscript.
