## [Peer Review File · Nature Communications]

Reviewers' comments:

Reviewer #1 (Remarks to the Author):

In this work, D'Amato et al., describe a poly(pro-drug) biomaterial scaffolds (electrospun fibers and films) that release E2, at a safe and efficacious dose, as they degrade slowly via hydrolysis, for a long period of time. Although an interesting work, this is more indicated for a more specialized journal, as I elucidate below.

The major new aspect of this paper is the synthesis of the poly(pro-drug) and the polymer, which could be more suitable for a chemical medicine journal. Electrospun fibers have been widely used before for many different applications. Since the idea of the work was to demonstrate the degradation and drug release kinetics for films and microfibers of P1 materials for a long period of time, authors have used only in vitro methods to access this, and even under non-physiological conditions (60 and 79 °C). It is justified that at 37 °C the degradation takes too long, however for a clinical translation this is a must and should be demonstrated in physiological conditions and also in a in vivo setup. Therefore, the very long release is merely based on in vitro predictions/extrapolation/assumptions, and not in real experimental demonstration, which makes the study rather limited and not convincing. In addition, there is no evidence in any in vivo model to confirm the in vitro claims, and thus, the E2 released during P1 degradation is neurotrophic and neuroprotective against oxidative stress in vitro is not significant to merit publication in Nature Communication.

Reviewer #2 (Remarks to the Author):

In the manuscript entitled "Vastly Extended Release of 17 β -Estradiol from Poly(pro-drug) Scaffolds for Neural Tissue Engineering" the authors have created a prodrug form of estradiol by polymerizing it through degradable carbamate bonds. Their intended use is for treatment of central nervous system injuries, an important area due to the lack of any available treatments that show appreciable improvement in human disease. They created these materials in films and fibers for in vitro release profiles of estradiol predicted to have a longevity of years. The authors have done a thorough job of material characterization and in vitro degradation/release kinetics and mechanism in a well-written manuscript. They have furthermore shown support that their materials stimulate neurite extension of primary neuronal cultures and protect cells from apoptosis due to oxidative stress, a destructive physiologic process in central nervous system injuries. A major limitation of the presented study is that the environment in living tissue was not fully simulated in the presented figures. In the injury, there is oxidative stress, reactive oxygen species, nutrient deprivation, changes in pH, and importantly infiltration of immune cells. The authors have recapitulated the oxidative stress, reactive oxygen species, and nutrient deprivation in their culture model, however how the non-neuronal cell component (immune cells, glia) of the injury contributes to the degradation profile of their material is unexplored in the current manuscript. Therefore, experiments to support extended degradation rates of their materials in the presence of cells that recapitulate the in vivo environment is requested. Other specific comments are below.

Specific comments

1. Figure 6d. Please indicate what the symbol * denotes and what statistical test was used to evaluate significance (the statistical test is indicated in the materials and methods, but it would be useful to have it available in the figure legend).
2. Figure 8d. The y-axis label "average neurite length" is misleading. Although it is stated in the text that it is the average of the ten longest neurites, the axis label could be improved for clarity. Please denote what the symbols * and # indicate and what statistical test was used to evaluate significance. The "SEM scale bar" is referred to, however, it is believed that there is no SEM in this particular figures, but visible wavelength microscopy.
3. Figure 9. Is there any effect of the scaffold itself or free estradiol in protecting neurons from

apoptosis?

4. Figure 9c. Please denote what the symbol * indicates and what statistical test was used.

It would be helpful to have context for how much E2 drug is released and how that is related to amounts delivered to show improvements in animal models of injury.

5. It would be helpful to the reader to have an analysis of therapeutic levels of E2 that are required in humans/animal models of disease and what total mass of implanted P1 would be required to sustain that therapeutic level.

Reviewer #3 (Remarks to the Author):

This work provided a novel strategy to realize long-term release of estradiol from electronspun prodrug polymers for treating central nervous system (CNS) injuries. Although the authors proposed a promising therapy for (CNS) injuries, more data are needed to demonstrate the significance of this work. The conclusions were overstated in several places. Major revision was recommended.

1. As mentioned in the instruction, "release E2 upon hydrolytic degradation over timescales of 1-10 years." But the drug release experiment is taken under 60 °C and 79 °C. The release behavior under 37 °C was calculated based on Arrhenius relation, this is not solid enough to endure the timescale of degradation under 37 °C. LC-MS should be used to monitor the cumulative release of estradiol.

2. The quantitative evaluation of degradation experiment was taken only in vitro in PBS buffer, and the degradation mechanism study was taken under the condition of neuron culture media, but the hydrolysis kinetics may be quite different when it's done in physical conditions. It is known that the degradation of polycarbonate can be significantly accelerated in vivo by macrophages (Bat, Erhan, et al. "Macrophage-mediated erosion of gamma irradiated poly (trimethylene carbonate) films." *Biomaterials* 30.22 (2009): 3652-3661.). Therefore, in vivo degradation behavior of poly(pro-drug) fibers must be examined.

3. All the neurotrophism and neuroprotection experiments were taken in vitro with neuron cells, when it goes to animal bodies, will it work as efficient as the in vitro results demonstrated? An animal model is needed to evaluate the potential for neural tissue engineering applications.

4. The material exhibits a fancy characteristic for neural tissue engineering, but what about the biocompatibility of this new material? It's good for neural cells, but what about other cells and tissue? Biocompatibility and safety evaluation should be carried out.

Reviewer #1 (Remarks to the Author):

In this work, D'Amato et al., describe a poly(pro-drug) biomaterial scaffolds (electrospun fibers and films) that release E2, at a safe and efficacious dose, as they degrade slowly via hydrolysis, for a long period of time. Although an interesting work, this is more indicated for a more specialized journal, as I elucidate below.

The major new aspect of this paper is the synthesis of the poly(pro-drug) and the polymer, which could be more suitable for a chemical medicine journal. Electrospun fibers have been widely used before for many different applications.

We thank this reviewer for reading and commenting on the manuscript. While it is indeed true that electrospun fibers have been used quite extensively (as we discussed and cited in the manuscript), there are several key novelties in this work of broad interest to the readership of *Nature Communications*:

- We demonstrate the first polycarbonate pro-drug formulation of E2 and successfully process this new polymer into bioactive films and electrospun fibers
- These materials display vastly extended E2 release in aqueous buffer and retain their robustness in the presence of activated macrophages
- Fiber scaffolds provide contact guidance *in vitro*
- E2 released during polymer degradation is neurotrophic and neuroprotective against oxidative stress

Since the idea of the work was to demonstrate the degradation and drug release kinetics for films and microfibers of P1 materials for a long period of time, authors have used only *in vitro* methods to access this, and even under non-physiological conditions (60 and 79 °C). It is justified that at 37 °C the degradation takes too long, however for a clinical translation this is a must and should be demonstrated in physiological conditions and also in a *in vivo* setup.

Therefore, the very long release is merely based on *in vitro* predictions/extrapolation/assumptions, and not in real experimental demonstration, which makes the study rather limited and not convincing.

It is important to emphasize that predictions from accelerated degradation tests were used only in the case of *fibers*, since the very small amount of material released per day at 37 °C was undetectable. However, in the case of thick films, there was no prediction/assumption used: we performed the degradation at 37 °C and were able to detect released E2 as a function of time simply because there is a much larger amount of material present in a film relative to a microfiber. Accordingly, we added a comment to clarify this matter. See **Page 8, line 4-6.**

Furthermore, to mimic the *in vivo* environment, we performed additional degradation experiments in the presence of activated macrophages (as suggested by another reviewer). We found that slow degradation of P1 films is not abrogated by culture conditions that mimic an inflammatory, physiologically mimetic environment. The prolonged degradation effect is still quite remarkable even when the scaffold is exposed to cells that recapitulate the *in vivo* environment, and are known to accelerate polymer degradation. This new experiment is

given as Supplementary Figure 10 and discussion is included in the manuscript (Page 9, line 15 – Page 10, line 16).

In addition, there is no evidence in any *in vivo* model to confirm the *in vitro* claims, and thus, the E2 released during P1 degradation is neurotrophic and neuroprotective against oxidative stress *in vitro* is not significant to merit publication in Nature Communication.

We understand and appreciate the need for assessment *in vivo*, for this new material, which is indeed a critical aspect of our ongoing work. In this first paper on the subject, we sought to evaluate the inherent physical properties and the *in vitro* performance of our highly novel biomaterials. The excellent results obtained strongly motivate us to pursue animal models, but these studies constitute the next phase of our research program and are outside the scope of the present manuscript. We added a few sentence in the introduction to clarify these points (Page 4, lines 17-22).

We also added several new sentences related to clinically used small molecule pro-drugs of E2, which are proven effective to prolong release to an extent (Page 15, Line 9 – Page 16, Line 5). Since our materials are high MW polymers composed of such pro-drugs, it is quite reasonable to speculate that these new materials will show vastly prolonged release in future animal studies. Our future work aims to bear this out in due course.

Reviewer #2 (Remarks to the Author):

In the manuscript entitled “Vastly Extended Release of 17 β -Estradiol from Poly(pro-drug) Scaffolds for Neural Tissue Engineering” the authors have created a prodrug form of estradiol by polymerizing it through degradable carbamate bonds. Their intended use is for treatment of central nervous system injuries, an important area due to the lack of any available treatments that show appreciable improvement in human disease. They created these materials in films and fibers for *in vitro* release profiles of estradiol predicted to have a longevity of years. The authors have done a thorough job of material characterization and *in vitro* degradation/release kinetics and mechanism in a well-written manuscript. They have furthermore shown support that their materials stimulate neurite extension of primary neuronal cultures and protect cells from apoptosis due to oxidative stress, a destructive physiologic process in central nervous system injuries.

We thank the reviewer for a thoughtful reading of our manuscript and for these generally quite positive and constructive comments. Detailed point-by-point responses are provided below.

A major limitation of the presented study is that the environment in living tissue was not fully simulated in the presented figures. In the injury, there is oxidative stress, reactive oxygen species, nutrient deprivation, changes in pH, and importantly infiltration of immune cells. The authors have recapitulated the oxidative stress, reactive oxygen species, and nutrient deprivation in their culture model, however how the ***non-neuronal cell component (immune cells, glia)*** of the injury contributes to the degradation profile of their material is unexplored in the current

manuscript. Therefore, experiments to support extended degradation rates of their materials in the presence of cells that *recapitulate the in vivo* environment is requested. Other specific comments are below.

These are excellent suggestions. Indeed, the response of these materials to non-neuronal cells is a critical consideration. Accordingly, we performed two additional experiments with astrocytes and macrophages. First, we performed an LDH assay with primary rat spinal cord astrocytes to demonstrate that P1 is not cytotoxic to these cells, as shown in Figure 10 (Page 14, lines 7-10). We also studied degradation in the presence of cells that recapitulate the *in vivo* environment. As also mentioned by reviewer #3, activated macrophages can have a strong impact on the degradation rates of polymers. In response, we performed assays with activated macrophages on films of the P1 material according to the reference provided by this reviewer. After 7 days, the mass and thickness of the P1 films cultured with activated macrophages did not significantly differ from those incubated with blank media nor from the untreated control samples. Thus, at least in the first week, we do not see a dramatic acceleration due to macrophages. Following these incubations, recovered polymer was analyzed again by GPC to assess any changes in molecular weight. Indeed, the sample incubated in media shows an identical chromatograph relative to untreated control. The sample exposed to activated macrophage did exhibit a slight shoulder peak on the lower MW side of the chromatograph, suggesting that macrophages do have some effect in terms of chain scission. The Supplementary Figure 10 shows all data related to accelerated degradation with macrophages. We have added a discussion in our revised manuscript (Page 9, line 15 – Page 10, line 16) and included the suggested reference [ref #45].

Specific comments

1. Figure 6d. Please indicate what the symbol * denotes and what statistical test was used to evaluate significance (the statistical test is indicated in the materials and methods, but it would be useful to have it available in the figure legend).

Thank you for the suggestion. This change has been made.

2. Figure 8d. The y-axis label “average neurite length” is misleading. Although it is stated in the text that it is the average of the ten longest neurites, the axis label could be improved for clarity.

Thank you for the suggestion. We changed this label to “Average *Maximum* Neurite Length” in Figure 8d.

Please denote what the symbols * and # indicate and what statistical test was used to evaluate significance.

Thank you for the suggestion. This change has been made (Figure 8 caption, page 12 and Figure 9, page 13).

The “SEM scale bar” is referred to, however, it is believed that there is no SEM in this particular figures, but visible wavelength microscopy.

Apologies for the typo; now corrected.

3. Figure 9. Is there any effect of the scaffold itself or free estradiol in protecting neurons from apoptosis?

Whether the polymeric form of E2 is capable of neuroprotection by some mechanism remains unknown, but we consider it unlikely since the hydroxy and phenoxy groups in E2, which are critical to its bioactivity, are both masked as carbonates in the polymer backbone. We included a sentence to this effect (see page 19, line 4-8).

4. Figure 9c. Please denote what the symbol * indicates and what statistical test was used. It would be helpful to have context for how much E2 drug is released and how that is related to amounts delivered to show improvements in animal models of injury.

Thank you for the suggestion. This change has been made.

5. It would be helpful to the reader to have an analysis of therapeutic levels of E2 that are required in humans/animal models of disease and what total mass of implanted P1 would be required to sustain that therapeutic level.

Thank you for the suggestion. We included some new sentences to this effect in the revised manuscript (see page 16, line 1-5).

Reviewer #3 (Remarks to the Author):

This work provided a novel strategy to realize long-term release of estradiol from electronspun prodrug polymers for treating central nervous system (CNS) injuries. Although the authors proposed a promising therapy for (CNS) injuries, more data are needed to demonstrate the significance of this work. The conclusions were overstated in several places. Major revision was recommended.

We thank the reviewer for a thoughtful reading of our manuscript and for these constructive comments. We have carefully revised our manuscript, including a title change, to be sure that we are not overstating the conclusions from any of the results in this study. Detailed point-by-point responses are provided below.

1. As mentioned in the instruction, ".....release E2 upon hydrolytic degradation over timescales of 1-10 years." But the drug release experiment is taken under 60 and 79 C. The release behavior under 37 C was calculated based on Arrhenius relation, this is not solid enough to endure the timescale of degradation under 37 C. LC-MS should be used to monitor the cumulative release of estradiol.

It is important to emphasize that predictions from accelerated degradation tests were used in the case of *fibers*, since the very small amount of material released per day at 37 °C was undetectable by fluorescence (and also by LCMS, which was attempted without success).

However, in the case of *thick films*, there was no prediction/assumption used: we performed the degradation at 37 °C and were able to detect released E2 as a function of time simply because there is a much larger amount of material present in a film relative to a microfiber. We added a sentence to this effect in the revised manuscript in order to clarify (Page 8, line 4-6).

2. The quantitative evaluation of degradation experiment was taken only *in vitro* in PBS buffer, and the degradation mechanism study was taken under the condition of neuron culture media, but the hydrolysis kinetics may be quite different when it's done in physical conditions. It is known that the degradation of polycarbonate can be significantly accelerated *in vivo* by macrophages (Bat, Erhan, et al. "Macrophage-mediated erosion of gamma irradiated poly (trimethylene carbonate) films." *Biomaterials* 30.22 (2009): 3652-3661.). Therefore, *in vivo* degradation behavior of poly(pro-drug) fibers must be examined.

Thank you for the excellent suggestion!

In response, we performed assays with activated macrophages on films of the **P1** material according to the reference provided by this reviewer (now included as reference #45). After 7 days, the mass and thickness of the films incubated with macrophages did not significantly differ from those incubated with blank media nor from the untreated control samples. Thus, at least in the first week, we do not see a dramatic acceleration due to macrophages. Following these incubations, recovered polymer was analyzed again by GPC to assess any changes in molecular weight. The sample incubated in media shows an identical chromatograph relative to untreated control. The sample exposed to activated macrophage did exhibit a slight shoulder peak on the lower MW side of the chromatograph, which slightly decreased the average MW (though not statistically significant change). This suggested that macrophages do have some effect in terms of accelerating chain scission at the molecular level, but this effect is not manifested as dramatically increased rate of material erosion at least at the 1-week time point. In all, these data suggest that the material indeed retain their slow degradation properties even in the presence of activated macrophages.

The Supplementary Figure 10 shows all data related to accelerated degradation with macrophages. We have added a discussion in our revised manuscript (Page 9, line 15 – Page 10, line 16) and included the suggested reference [ref #45].

3. All the neurotrophism and neuroprotection experiments were taken *in vitro* with neuron cells, when it goes to animal bodies, will it work as efficient as the *in vitro* results demonstrated? An animal model is needed to evaluate the potential for neural tissue engineering applications.

Activity of biomaterials may indeed differ between *in vitro* and *in vivo* environments. The *in vitro* data here was viewed as a necessary first step, due to the fact that we are exploring

highly novel biomaterials that were newly synthesized in our lab, with rather unique chemical structure and physical properties. In the revised manuscript, we took care to avoid any misrepresentation of the fact that this initial study is focused entirely on *in vitro* degradation. The animal models needed to assess *in vivo* degradation will be the subject of our future work.

Furthermore, we included additional references and discussions regarding pro-drugs of estrogen that are **currently in clinical use** specifically for extended release application, in order to highlight the fact that this class of materials is highly likely to extend release in the *in vivo* context based on what is known about the small molecule pro-drugs (page 9, line 15 – page 10, line 16).

4. The material exhibits a fancy characteristic for neural tissue engineering, but what about the **biocompatibility** of this new material? It's good for neural cells, but what about other cells and tissue? Biocompatibility and safety evaluation should be carried out.

Thank you for raising this key point. The compounds released from this material upon degradation are generally considered non-toxic and biocompatible at the relevant concentrations: PEG, CO₂, and estrogen. To confirm that the **P1** materials are indeed non-toxic, we performed an LDH assay to quantify the cytotoxicity against astrocytes and found no significantly adverse effects, as we expected. Please see new **Figure 10** and discussion in text (page 13, line 15 – page 14, line 5). In our future work, safety assessments upon implantation in live animals will be performed in the future and reported in due course.

EDITORIAL NOTE:

As reviewer #1 was unable to review the revision this round they were replaced with reviewer #4

As reviewer #3 was unable to review the revision this round they were replaced with reviewer #5

REVIEWERS' COMMENTS:

Reviewer #2 (Remarks to the Author):

In response to comment 5, the authors have included the discussion: "Pharmaceutical applications of E2 in the clinic typically increase 1 serum E2 concentration by 10-150 pg/mL^{49,53}. One of the major breakthroughs herein is that these novel materials can deliver from ~50 ng E2/day (fibers) to 260 ng E2/day (films), for orders of magnitude longer timescales than reported previously in the literature. By tailoring the amount of material applied, E2 serum concentrations on the order of pg/mL could be readily attainable." After examining the methods, it is difficult to determine how much total polymeric material would be needed to achieve the stated therapeutic levels of E2. It is requested that the authors state more explicitly how "By tailoring the amount of material applied, E2 serum concentrations on the order of pg/mL could be readily attainable." to make clear the theoretical total mass of materials that would be needed to be implanted in humans in order to achieve therapeutic effects. Is this total mass a reasonable amount to implant?

The other comments have been answered satisfactorily.

Reviewer #4 (Remarks to the Author):

Nature Commun review

I reviewed the manuscript and author's response to the reviewers. Specifically for Reviewer #1, in my opinion:

1. I agree with the author: this work is novel in it's the first pro-drug polymer of estradiol, which can be electrospun into fibers, and then displays impressive biological behavior.
2. The accelerated in vitro studies are appropriate, particularly given the scale of fibers (vs. thick films. Further, the inclusion of degradation in the presence of activated macrophages clearly demonstrates the ability of these novel materials to mitigate inflammation.
3. I agree that in vivo studies will be more extensive and that this work stands alone as the first – as outlined in my first comment.

Reviewer #5 (Remarks to the Author):

In this revision, the authors have addressed the major concerns expressed by Reviewer 3. Specifically: (1) they have added assays with activated macrophages to better simulate the in vivo environment and (2) they have revised wording throughout to avoid implying that the work represents in vivo performance.

Whether these materials will perform in vivo remains unknown, however, as noted by several of the reviewers and acknowledged by the authors themselves. While it is appreciated that in vivo studies are out-of-scope for the present work, the impact is nevertheless reduced by the absence of in vivo proof-of-concept.

Point-by-point reply to comments

Text in black is copied from Editor's letter; Blue indented text is our reply

Your manuscript entitled "Vastly Extended Drug Release from Poly(pro-17 β -Estradiol) Materials Facilitates in vitro Neurotrophism and Neuroprotection" has now been seen again by our referees, whose comments appear below. In light of their advice I am delighted to say that we are happy, in principle, to publish a suitably revised version in Nature Communications under the open access CC BY license (Creative Commons Attribution v4.0 International License). We therefore invite you to revise your paper one last time to address the remaining concerns of our reviewers. At the same time we ask that you edit your manuscript to comply with our format requirements and to maximise the accessibility and therefore the impact of your work.

Thank you for consideration of this work. We have carefully read all the editor's and reviewer's comments and have edited the manuscript accordingly.

As you can see from the reports, reviewers #2 and #4 still have a couple of ongoing questions but we feel that they could be addressed by incorporating some additional discussion into the manuscript. We are confident that we should be able to assess the response in-house but we reserve the right to contact the reviewer again if we do not think that these requests have been fully addressed.

Unfortunately reviewer #3 was unable to look on the PBP response again and therefore we asked reviewer 4 to comment on the response given to reviewer #3.

We thank these reviewers for taking the time to read and critique our manuscript. Our responses to these helpful reviewer comments are found on pages 6 and 7 herein.

You might also consider to rephrase the sentence "To our knowledge, however, no polyprodrug has been processed into biomaterial scaffolds, such as electrospun fibers, that can promote, protect and guide the growth of neurons." on page 2, as there are examples in the literature (e.g. Griffin, J et al. J Biomed Mater Res: Part A, 97 (3) 230-242 (2011));

Thank you for the excellent suggestion. We have revised and moderated this discussion to accurately reflect the differences between Griffin et al (2011) and the present work. Please see new discussion on **page 2 line 21** through **page 3 line 3**.